# Effect of Phosphorus Precursor, Reduction Temperature, and Support on the Catalytic Properties of Nickel Phosphide Catalysts in Continuous-Flow Reductive Amination of Ethyl Levulinate

**DOI:** 10.3390/ijms23031106

**Published:** 2022-01-20

**Authors:** Yazhou Wang, Alexey L. Nuzhdin, Ivan V. Shamanaev, Evgeny G. Kodenev, Evgeny Yu. Gerasimov, Marina V. Bukhtiyarova, Galina A. Bukhtiyarova

**Affiliations:** 1Faculty of Natural Sciences, Novosibirsk State University, 630090 Novosibirsk, Russia; wangyazhou@yandex.ru; 2Boreskov Institute of Catalysis SB RAS, 630090 Novosibirsk, Russia; i.v.shamanaev@catalysis.ru (I.V.S.); kodenev_e@mail.ru (E.G.K.); gerasimov@catalysis.ru (E.Y.G.); mvb@catalysis.ru (M.V.B.); gab@catalysis.ru (G.A.B.)

**Keywords:** nickel phosphide, reductive amination, ethyl levulinate, *N*-alkyl-5-methyl-2-pyrrolidone, flow reactor, molecular hydrogen, support effect, reduction temperature, phosphorus precursor

## Abstract

Levulinic acid and its esters (e.g., ethyl levulinate, EL) are platform chemicals derived from biomass feedstocks that can be converted to a variety of valuable compounds. Reductive amination of levulinates with primary amines and H_2_ over heterogeneous catalysts is an attractive method for the synthesis of *N*-alkyl-5-methyl-2-pyrrolidones, which are an environmentally friendly alternative to the common solvent *N*-methyl-2-pyrrolidone (NMP). In the present work, the catalytic properties of the different nickel phosphide catalysts supported on SiO_2_ and Al_2_O_3_ were studied in a reductive amination of EL with *n*-hexylamine to *N*-hexyl-5-methyl-2-pyrrolidone (HMP) in a flow reactor. The influence of the phosphorus precursor, reduction temperature, reactant ratio, and addition of acidic diluters on the catalyst performance was investigated. The Ni_2_P/SiO_2_ catalyst prepared using (NH_4_)_2_HPO_4_ and reduced at 600 °C provides the highest HMP yield, which reaches 98%. Although the presence of acid sites and a sufficient hydrogenating ability are important factors determining the pyrrolidone yield, the selectivity also depends on the specific features of EL adsorption on active catalytic sites.

## 1. Introduction

Levulinic acid (LA) is a platform chemical derived from lignocellulose biomass with great potential to produce a wide range of valuable chemicals [1,2,3,4,5]. Levulinate esters (e.g., ethyl levulinate, EL) are considered as an alternative to LA due to their specific physicochemical properties [6,7,8]. Unlike LA, its esters do not corrode the equipment, dissolve well in non-polar organic solvents, and generally do not leach metals from the catalysts.

Reductive amination of LA or levulinates with primary amines over heterogeneous metal catalysts using molecular hydrogen as a reducing agent is a promising method for the synthesis of *N*-substituted-5-methyl-2-pyrrolidones [9,10,11,12,13,14,15,16,17,18,19,20,21,22,23,24], which are an alternative to the carcinogenic solvent *N*-methyl-2-pyrrolidone (NMP) used in large volumes in industry [3]. Due to the high cost and limited availability of precious metals, the use of non-noble metal catalysts is of particular interest [1,20,21,22,23,24,25].

In the case of EL, the process begins with the condensation of EL and amine to the corresponding imine, which reacts with hydrogen over metal sites to form 4-aminopentanoate, and further intramolecular amidation leads to pyrrolidone. The conversion of imine can also occur via imine-enamine equilibrium, followed by the elimination of EtOH and hydrogenation to the desired product [9,13,17,24]. At the same time, hydrogenation of EL with the subsequent cyclization gives γ-valerolactone (GVL) in a side reaction (Figure 1).

A positive role of acid sites on the surface of the support material in achieving a high pyrrolidone yield was observed, which is associated with the acceleration of EL amination and cyclization steps [9,10,11,12,13,14,21,25]. Nickel phosphides are attracting increased attention as bifunctional catalysts due to the presence of both metal and acid sites [26,27,28]. It is generally accepted that nickel phosphides contain weak Brønsted and Lewis acid sites, which are associated with P–OH groups and coordinatively unsaturated Ni^δ+^ sites, respectively. The acidity of Ni-phosphide catalysts depends on the reduction temperature, phosphorus precursor, support nature, and Ni:P ratio. The variation of these parameters significantly changes the activity of supported nickel phosphides in hydrodeoxygenation [29,30,31,32] and hydrodesulfurization [33] reactions. In our previous work, it was shown that 6.3% of the Ni_2_P/SiO_2_ catalyst prepared by impregnation of silica with aqueous solutions of Ni(CH_3_COO)_2_ and (NH_4_)_2_HPO_4_ provides continuous-flow reductive amination of EL to *N*-alkyl-5-methyl-2-pyrrolidones with the yield up to 94% [24]. Isolation of the target product from the reaction medium is a difficult task, the solution of which is facilitated by an increase in the yield.

The aim of this work is to determine the influence of the phosphorus precursor, reduction temperature, reactant ratio, and support nature on the catalytic properties of nickel phosphide catalysts in the reaction of EL with *n*-hexylamine (HA) and H_2_ in a flow reactor. The behavior of the Ni_2_P/SiO_2_ catalyst also has been studied in a physical mixture with acidic materials (γ-Al_2_O_3_, SAPO-11, zeolite β) to test whether the addition of such materials could improve the catalytic properties of Ni_2_P/SiO_2_ in the synthesis of *N*-hexyl-5-methyl-2-pyrrolidone.

## 2. Results and Discussion

### 2.1. Catalyst Characterization

A series of Ni_2_P/SiO_2_ catalysts was prepared by impregnation of SiO_2_ with aqueous solutions of Ni(CH_3_COO)_2_ and (NH_4_)_2_HPO_4_ or Ni(OH)_2_ and H_3_PO_3_ followed by an in situ, temperature-programmed reduction (TPR) in a hydrogen flow [29]. The catalysts were denoted by the letter “A” and “I” with respect to the P-containing precursor used: phosphate (A) and phosphite (I). The samples reduced at 450, 500, 550, and 600 °C were labeled as Ni_2_P/SiO_2__A(I)450, Ni_2_P/SiO_2__A(I)500, Ni_2_P/SiO_2__A(I)550, and Ni_2_P/SiO_2__A(I)600, respectively. Ni_2_P/Al_2_O_3_ catalysts were obtained starting from Ni(OH)_2_ and H_3_PO_3_ with subsequent in situ TPR at 550 °C (Ni_2_P/Al_2_O_3__550) and 600 °C (Ni_2_P/Al_2_O_3__600) [30]. To prepare the Ni_2_P phase, the impregnating solutions with an initial Ni/P molar ratio of 1/2 were used [29,30,31]. In addition, Ni/SiO_2_ and Ni/Al_2_O_3_ reference samples were prepared by impregnation of the support with an aqueous solution of Ni(CH_3_COO)_2_ followed by drying, calcination, and reduction in H_2_ flow at 400 °C [29]. The list and physicochemical properties of the catalysts used in the present study are shown in Table 1.

The prepared samples contain approximately the same amount of nickel (6.2–7.3 wt %) after ex situ reduction at corresponding temperature for 1 h. Ni_2_P/SiO_2__I samples contain higher amounts of phosphorus than Ni_2_P/SiO_2__A materials (Table 1). An increase in the reduction temperature leads to a decrease in the P content due to the formation of volatile compounds (PH_3_, P, P_2_, etc.) during reduction [29,30,31]. At the same time, the phosphorus content in the Ni_2_P/Al_2_O_3_ samples is significantly higher than in the Ni_2_P/SiO_2_ samples, which is explained by the formation of aluminum phosphates on the catalyst surface [30].

XRD patterns of some nickel phosphide catalysts are presented in Figure 1. All XRD curves show the characteristic signals of the Ni_2_P phase: 2θ = 40.7°, 44.5°, 47.3°, 54.1°, and 55.0° (a = b = 0.5859 nm, c = 0.3382 nm, α = β = 90°, γ = 120°; JCPDS #03-0953). In addition, the Ni_2_P/SiO_2_ and Ni_2_P/Al_2_O_3_ samples contain diffraction peaks of the support: the broad line at 2θ~15–30 from the amorphous SiO_2_ or characteristic peaks from γ-Al_2_O_3_ (PDF No. 29-0063). The average crystallite size of the Ni_2_P particles (D_XRD_) estimated using the Scherrer equation is 10, 4, and 4 nm in Ni_2_P/SiO_2__A600, Ni_2_P/SiO_2__I550, and Ni_2_P/Al_2_O_3__550, respectively.

According to TEM data, the Ni_2_P/SiO_2__A600 sample contains nickel phosphide particles with a mean particle size (*D_TEM_*) of 8.9 nm (Figure 2). The *D_TEM_* of Ni_2_P nanoparticles in Ni_2_P/SiO_2__I450, Ni_2_P/SiO_2__I500, and Ni_2_P/SiO_2__I550 samples is 1.8, 3.0, and 3.2 nm, respectively. Therefore, the TEM results show that the use of H_3_PO_3_ as a phosphorus precursor promotes the formation of smaller Ni_2_P nanoparticles [29].

The mean Ni_2_P particle diameters of the Ni_2_P/Al_2_O_3__550 and Ni_2_P/Al_2_O_3__600 samples are 2.8 and 3.1 nm, respectively [30]. Ni/Al_2_O_3_ contains 2–10 nm nanoparticles with nickel in the oxidized state. At the same time, TEM data of Ni/SiO_2_ sample show much larger particles of 5–50 nm in diameter.

Figure 3a shows the NH_3_-TPD profiles of Ni_2_P/SiO_2_ samples and SiO_2_ support. All samples have a signal with *T_max_* at 231–250 °C corresponding to weak acid sites. The total number of acid sites, estimated by integrating the NH_3_ desorption peaks, is presented in Table 1. The Ni_2_P/SiO_2__A600, Ni_2_P/SiO_2__I450, and Ni_2_P/SiO_2__I500 samples have a significantly higher quantity of acid sites compared to the SiO_2_ support (84 μmol g^–1^). Total acidity for Ni_2_P/SiO_2__I samples decreases with an increase in the reduction temperature from 450 to 550 °C, which is accompanied by a decrease in the amount of P–OH surface groups [29].

On the NH_3_-TPD curve of γ-alumina, two desorption peaks of ammonia centered at 237 and 335 °C were observed (Figure 3b). The first peak around 237 °C is attributed to the sites with the weakest acidity responsible for physisorbed and chemisorbed NH_3_, while the second peak at 335 °C belongs to the moderate-strength acid sites [30,34]. The total acidity of γ-Al_2_O_3_ support is 421 μmol g^–1^. The Ni_2_P/Al_2_O_3_ samples contain both weak and medium acid sites. With the increase in the reduction temperature from 550 to 600 °C, the total acidity of the catalyst is decreased from 477 to 354 μmol g^–1^. The amount of weak acid sites increases as compared to the Al_2_O_3_ support, while the number of the moderate-strength acid sites decreases. The latter can be explained by the shielding of the alumina surface by an excess of P-containing species [30].

### 2.2. Catalytic Activity

The catalytic properties of nickel phosphide catalysts were investigated in the continuous-flow reductive amination of EL with HA at 160–180 °C and total pressure of 10 bar using toluene as a solvent. Before each catalytic run, a fresh portion of the precursor was reduced in situ in hydrogen flow. It was found that *N*-hexyl-5-methyl-2-pyrrolidone (HMP) formed as the main product in the presence of all catalysts. In addition to HMP, γ-valerolactone (GVL), unsaturated *N*-hexyl-5-methyl-2-pyrrolidones (UHPs), ethanol, and dihexylamine were observed among the reaction products.

The Ni_2_P/SiO_2__A600 catalyst shows the formation of HMP with 96% selectivity at 98% conversion of EL [24]. A decrease in the reduction temperature to 500–550 °C leads to a decrease in the hydrogenation activity that, in turn, gives a lower yield of HMP (Table 2, entries 1−4). Apparently, this is due to the incomplete reduction in of phosphate groups at temperatures below 600 °C [29]. Therefore, the reduction temperature of 600 °C is optimal for the Ni_2_P/SiO_2__A sample because it allows forming a greater amount of Ni_2_P phase (responsible for hydrogenation reactions) along with the maintenance of a certain number of acid sites (responsible for imine formation and intramolecular amidation).

The Ni_2_P/SiO_2__I450 catalyst prepared using H_3_PO_3_ and reduced at 450 °C demonstrates HMP selectivity comparable to the Ni_2_P/SiO_2__A600, but the hydrogenation activity was low (Table 2, entries 5 and 6). To increase the EL conversion, the contact time was increased by reducing the liquid flow rate and increasing the catalyst loading (Table 2, entry 7). As a result, the HMP yield reached 93%. An increase in the reduction temperature to 500–550 °C leads to an increase in activity; however, it is accompanied by a decrease in the selectivity of HMP (Table 2, entries 8−11). The growth of the hydrogenation capacity in the series: Ni_2_P/SiO_2__I450 < Ni_2_P/SiO_2__I500 < Ni_2_P/SiO_2__I550 is probably associated with the formation of a larger number of Ni_2_P particles. However, a reduction in acid sites’ concentration in this order (Table 1) leads to a decrease in the pyrrolidone yield. Therefore, the higher selectivity of the Ni_2_P/SiO_2__I450 catalyst in comparison with the Ni_2_P/SiO_2__I500 or the Ni_2_P/SiO_2__I550 catalysts is probably explained by the higher amount of P–OH surface groups, which promotes condensation of EL with amine, preventing GVL formation.

Raising the temperature from 170 to 180 °C in the presence of the Ni_2_P/SiO_2__I550 catalyst reduces the HMP yield by increasing the rate of EL hydrogenation to GVL (Table 2, entries 10 and 11). However, in the case of the Ni_2_P/SiO_2__A500 and Ni_2_P/SiO_2__I450 samples with lower hydrogenation activity, increasing the temperature to 180 °C shows an increase in HMP yield (Table 2, entries 3−6) due to both an increase in EL conversion and a decrease in selectivity to UHPs.

The effect of the support nature (SiO_2_ or γ-Al_2_O_3_) on the catalytic properties of nickel phosphide catalysts in the reductive amination of EL was also considered. In the case of Ni_2_P/Al_2_O_3__550 and Ni_2_P/Al_2_O_3__600 samples, HMP yield is lower than for Ni_2_P/SiO_2__A600 (Table 2, entries 12−14) due to hydrogenation of EL to GVL and 1,4-pentanediol (a product of GVL hydrogenation). The noticeably lower yield of the target product on the Ni_2_P/Al_2_O_3_ catalysts, which are not inferior to Ni_2_P/SiO_2__A600 in terms of hydrogenation ability and concentration of acid sites, indicates that the reaction selectivity depends on the specific adsorption of the substrates on active sites. In the presence of Al_2_O_3_-supported catalysts, EL is probably adsorbed through the carbonyl group on the Lewis sites of γ-Al_2_O_3_ [35,36] that increases the rate of EL hydrogenation and, accordingly, decreases the selectivity to HMP. Since the spillover of H atoms to the non-reducible supports, such as γ-Al_2_O_3_ and SiO_2_, is practically impossible [37], hydrogenation probably occurs at the border between the Ni_2_P nanoparticles and support.

It should be noted that the Ni/Al_2_O_3_ catalyst has a high hydrogenation activity (the full conversion of EL is observed already at 150 °C), but the selectivity to HMP is very low due to the high hydrogenation rate of EL to GVL and 1,4-pentanediol (Table 2, entry 15). At the same time, the Ni/SiO_2_ catalyst provides a similar EL conversion as compared with Ni_2_P/SiO_2__A600. However, the HMP selectivity was noticeably lower (Table 2, entry 16). Thus, the Ni_2_P/SiO_2__A600 sample provides the maximum yield of HMP among all investigated nickel catalysts, and the following experiments were carried out using this catalyst.

The formation of dihexylamine during the reaction is probably associated with the condensation of *n*-hexylamine molecules on the catalyst surface in the presence of hydrogen. This reaction is competitive with the imine formation, which leads to a decrease in the yield of HMP at a ratio EL/HA~1. The slight excess of HA at the ratio of EL/HA equal to 1:1.2 resulted in the increase in HMP yield to 98% (Table 2, entry 17). The Ni_2_P/SiO_2__A600 catalyst shows good stability under these reaction conditions. The time-dependent study of the reductive amination of EL with HA demonstrates that the EL conversion and HMP yield remained unchanged for 6 h (Figure 4).

In previous studies, we found a synergetic effect of Ni_2_P/SiO_2_ and γ-Al_2_O_3_ in hydrodeoxygenation of methyl palmitate, which was explained by the cooperation of the metal sites of Ni_2_P/SiO_2_ and the acid sites of γ-alumina for metal-catalyzed and acid-catalyzed reactions [34]. In this work, catalytic properties of physical mixtures of the Ni_2_P/SiO_2__A600 catalyst with γ-Al_2_O_3_ and other acidic diluters (SAPO-11 and zeolite β) were investigated in the reductive amination of EL with HA. The physicochemical properties of the diluters are shown in Appendix A. According to NH_3_-TPD data, the acidity of the diluters is decreased in the following order: zeolite β (1920 μmol g^–1^) > SAPO-11 (1110 μmol g^–1^) > γ-Al_2_O_3_ (421 μmol g^–1^)

It was found that mixing the Ni_2_P/SiO_2__A600 catalyst with γ-alumina at a ratio of 1:1 increases the EL conversion due to an increase in the rates of amination and cyclization reactions on acid sites of γ-Al_2_O_3_; however, the HMP selectivity does not change (Table 3, entries 1−3). When alumina was placed at the reactor inlet separately from the phosphide catalyst, the material balance decreased to ~90%, which is associated with the formation of high-molecular-weight by-products on Al_2_O_3_ acid sites. At the same time, the use of physical mixtures of Ni_2_P/SiO_2__A600 with SAPO-11 or zeolite β has practically no effect on the HMP yield (Table 3, entries 4−6) that can be explained by the strong chemisorption of amine on the Brønsted acid sites of these diluters.

## 3. Conclusions

The catalytic properties of the supported nickel phosphide catalysts differing in preparation conditions were studied in the continuous-flow reductive amination of ethyl levulinate (EL) with *n*-hexylamine (HA) to *N*-hexyl-5-methyl-2-pyrrolidone (HMP). The Ni_2_P/SiO_2__A600 catalyst prepared using (NH_4_)_2_HPO_4_ and reduced at 600 °C provides the highest HMP yield, which reaches 98% when using a 20% excess of amine. A decrease in the reduction temperature below 600 °C gives a lower yield of HMP due to incomplete reduction of phosphate groups to Ni_2_P nanoparticles. The catalysts obtained starting from H_3_PO_3_ show a lower yield of the target product than the Ni_2_P/SiO_2__A600 under the same conditions. In the case of the Ni_2_P/Al_2_O_3_ samples, which are not inferior to Ni_2_P/SiO_2__A600 in hydrogenation activity and concentration of acid sites, the HMP yield was lower due to EL hydrogenation in a side reaction. Thus, although the presence of acid sites and a sufficient hydrogenating ability are important factors determining the pyrrolidone yield, the selectivity also depends on the specific features of EL adsorption on the active catalytic sites. Further studies should be directed toward the creation of bifunctional catalysts, in the presence of which the adsorption of EL through carbonyl group on metal sites is minimized, which prevents its hydrogenation.

## 4. Materials and Methods

### 4.1. Chemicals

Ethyl levulinate (98%, Acros Organics), *n*-hexylamine (99%, Acros Organics), *n*-decane (99%, Acros Organics), and toluene (99.5%, “ECOS” Russia) were used without additional purification. To prepare the catalysts, Ni(OH)_2_ (≥98%, Acros Organics), Ni(CH_3_COO)_2_·4H_2_O (≥98%, Reachim, Russia), H_3_PO_3_ (≥97%, Sigma-Aldrich), and (NH_4_)_2_HPO_4_ (Alfa Aesar, technical grade) were used. The silica (“KSKG”) and γ-alumina (“IKGO-1”) were supplied from “ChromAnalit” Ltd. (Moscow, Russia) and “Promkataliz” Ltd. (Ryazan, Russia), respectively. Commercial SiC (Chelyabinsk Plant of Abrasive Materials, Chelyabinsk, Russia), zeolite β in H form (Angarsk catalyst and organic synthesis plant, Angarsk, Russia), and SAPO-11 (Zeolyst International, Conshohocken, PA, USA) were utilized as diluters.

### 4.2. Catalyst Preparation

The catalysts were prepared by impregnation of the support (fraction of 0.25–0.50 mm) with aqueous solutions of Ni and P precursors (the initial Ni/P molar ratio of 0.5) [29,30,31]. 

*Phosphate method* (Ni_2_P/SiO_2__A). Ni(CH_3_COO)_2_·4H_2_O (1 eqv.) was added to an aqueous solution of (NH_4_)_2_HPO_4_ (2 eqv.) with stirring to form a yellow–green precipitate. Afterward, concentrated HNO_3_ was added dropwise to dissolve the precipitate, and SiO_2_ support was impregnated by the obtained solution. The precursors were dried in air at room temperature overnight, at 110 °C, and calcined at 500 °C for 3 h [29].

*Phosphite method* (Ni_2_P/SiO_2__I, Ni_2_P/Al_2_O_3_). Ni(OH)_2_ (1 eqv.) was added to an aqueous solution of H_3_PO_3_ (2 eqv.) with stirring. Granules of SiO_2_ or γ-Al_2_O_3_ were impregnated by the obtained solution. The precursors were dried at room temperature in air overnight and at 80 °C for 24 h [29,30].

To compare phosphide and metal catalysts, Ni/SiO_2_ and Ni/Al_2_O_3_ reference samples were prepared by the impregnation of the support with an aqueous solution of Ni(CH_3_COO)_2_ followed by drying and calcination at 500 °C for 3 h [29].

### 4.3. Catalyst Characterization

The Ni and P content were determined by atomic absorption spectroscopy using an Optima 4300 DV analyzer (Perkin Elmer, Waltham, MA, USA). The TEM studies were performed on a JEM-2010 electron microscope (JEOL, Tokyo, Japan). Powder XRD patterns were recorded on a Bruker D8 Advance diffractometer (Bruker, Billerica, MA, USA) using CuK_a_ radiation. The acidic properties of the catalysts were investigated by temperature-programmed desorption of ammonia (NH_3_-TPD) using an Autosorb-1 instrument (Quantachrome Instruments, Boynton Beach, FL, USA) [29,30,34]. Textural characteristics were obtained from N_2_ adsorption–desorption isotherms measured at 77 K on a Micromeritics ASAP^®^ 2400 device (Micromeritics, Norcross, GA, USA).

### 4.4. Catalyst Performance

The investigation of the catalytic properties in the reductive amination of EL with *n*-hexylamine was performed using fixed-bed flow reactor (inner diameter of 9 mm, length of 265 mm). The catalyst precursor (750 mg) was diluted with silicon carbide (fraction of 0.25–0.50 mm) or a mixture of SiC with another diluter (γ-Al_2_O_3_, SAPO-11, and zeolite β) and placed in the reactor between two SiC layers. Before the experiments, the precursor was reduced in situ in a hydrogen flow (100 mL min^−1^) at atmospheric pressure. The samples were heated to 450–600 °C (for Ni_2_P/SiO_2_ and Ni_2_P/Al_2_O_3_) or to 400 °C (for Ni/SiO_2_ and Ni/Al_2_O_3_) at a heating rate of 1 °C min^−1^ and kept at the reduction temperature for 1 h or 2 h, respectively [24,29,30,31,34].

After pre-reduction of the catalyst, the temperature was reduced to 170 °C, and toluene was pumped through the flow reactor. Afterward, the inlet was switched to the flask containing the reaction mixture, and this point in time was chosen as the starting point of the reaction. In the standard experiment, the solution of EL (0.04 M) and HA (0.041 M) in toluene was used with *n*-decane as the internal standard. The reaction was carried out in the 150–180 °C range, 10 bar total pressure, where liquid and hydrogen flow rates were set to 0.33 and 30 mL min^−^^1^, respectively. The catalyst performance was assessed by averaging three samples taken in the intervals of 2.5–3, 3–3.5, and 3.5–4 h from the beginning of the catalytic test.

The reaction products were analyzed by GC (Agilent 6890N instrument with an HP 1-MS capillary column). The conversion, selectivity, and yield were calculated based on EL. The reaction products were identified by GC–MS. The material balance between the inlet and outlet streams usually exceeded 98% [24].

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
