# Peer review of "Effect of Phosphorus Precursor, Reduction Temperature, and Support on the Catalytic Properties of Nickel Phosphide Catalysts in Continuous-Flow Reductive Amination of Ethyl Levulinate"

_ijms, 2022, doi:10.3390/ijms23031106_

Round 1

Reviewer 1 Report

The authors reported in the manuscript the development nickel phosphide immobilized onto the solid supports SiO2 or Al2O3 to act as heterogeneous catalyst for the reductive amination of ethyl levuninate to N-hexyl-5-methyl-2-pyrrolidone.

Overall, the manuscript is very well written, with a very detailed experimental part and the materials characterization as well as a strong discussion of the obtained results of the heterogeneous catalysis.

Nevertheless, minor revisions and modifications are necessary:

Page 3 line 92: “phosphorus content in Ni2P/Al2O3 samples is significantly higher than in Ni2P/SiO2, which is explained by the formation of aluminum phosphates on the catalyst surface”. This evidence can be investigated using the characterization technique X-ray photoelectron spectroscopy (XPS), by the deconvolution of phosphor and/or aluminium high-resolution spectra. If possible include this characterization in the manuscript.

In the XRD characterization, is possible to determine the average crystallite size using the Debye-Scherrer equation as well the lattice parameter. Please include this information in the manuscript.

Page 9 line 251: “Ni/P molar ratio of 0.5” this molar ratio corresponds to P/Ni and not Ni/P, please correct

After these modifications, I consider the article suitable to be published in International Journal of Molecular Sciences.

Reviewer 2 Report

Effect of phosphorous precursor, reduction temperature and support on the catalytic properties of nickel phosphide catalysts in continuous-flow reductive amination of ethyl levulinate

Comment to the Authors

In this interesting article authors Yazhou Wang, Alexey L. Nuzhdin, Ivan V. Shamanaev, Evgeny G. Kodenev, Evgeny Yu. Gerasimov, Marina V. Bukhtiyarova and Galina A. Bukhtiyarova discussed catalytic properties of nikel phosphide catalysts with SiO2 and Al2O3 were investigated in reductive amination of ethyl levulinate. Ethyl levulinate with similar physicochemical properties, non-erosive property, better solubility in organic solvents and ability to not leaching metals from the catalysts make them better replacements to levulinic acids, which are basis of variety of chemicals synthesis. Effect of phosphorus precursor, reduction temperature, reactant ratio and addition of acidic diluters on the catalyst performance was also studied.

Authors haven’t clearly discussed novelty of this study. Such discussion will help improve significance of this study. Authors also haven’t discussed the significance of selection of using parameters like influence of phosphorous precursor, reduction temperature, reaction ratio and nature of catalyst in this study. Why only certain physiochemical properties were used to study properties of the catalyst. Including such discussion will make this article more helpful and interesting for the broad readers interested in this field. It would be also helpful for the readers if authors can add some comments for the reason of using specific reference samples (for e.g. Ni/Al2O3 and Ni(CH3COO)2) and their role while testing their hypothesis.

Overall, this is interesting study in the field of chemical and organic synthesis, and catalysis. Findings provided are reasonable and backed up by experimental results but need further discussion to make it easier for the readers. The following points needs to be addressed before publication, to make this article more helpful for the readers of International Journal of Molecular Sciences:

  1. It would be great if authors can discuss more and explain the significance of the obtained results. Why particle size of Ni2P/SiO2_A500 and Ni2P/SiO2_A550 was not determined. It will also make it easier to readers if authors define short forms, they used at the end of the table for e.g., for “n.d.”.
  2. Authors mentioned, decrease in reduction temperature decreases hydrogenation activity, and therefore lowers HMP yield. What affects GVL and UHPs selectivity? Authors comment on it would be helpful to understand in future studies to reduce their selectivity to increase the HMP yield.
  3. It would be helpful if authors can comment on their finding for (entry 10 and 11) in Table 2: Why for Ni2P/SiO2_I550 (entry 10 and 11) with increase in temperature from 170 to 180 ⁰C increases conversion of EL (from 92 to 97%), however, it decreases HMP selectivity (from 96 to 87%) and yield (from 88 to 84%)? i.e. increase in temperature from 170 to 180 ⁰C for Ni2P/SiO2_A500 (entries 3 and 4) and Ni2P/SiO2_I450 (entries 5 and 6) shows increase in conversion of EL, decrease selectivity of UHPs (slight difference in GVL selectivity) but increase in HMP selectivity and percent yield. What could be the reason for such trend change for Ni2P/SiO2_I550? Was temperature effect on Ni2P/SiO2_A600 studied? If no, why not and what results authors would predict?
  4. It would be also helpful if authors provide few lines on their recommendations for fine tuning catalysts to further increase their activity for future studies.

Minor corrections:

  1. Page 4 line 109: Figure number is incorrect. It should be Figure 2 instead 3 is written.
  2. Page 8 line 228: Typo in Nanoparticles spelling: “nanoparticales” is written
